# Engineering stringent genetic biocontainment of yeast with a protein stability switch

Stefan A. Hoffmann [1] & Yizhi Cai [1] ✉

Synthetic biology holds immense promise to tackle key problems in resource use, environmental remediation, and human health care. However, comprehensive safety measures are lacking to employ engineered microorganisms in open-environment applications. Genetically encoded biocontainment systems may solve this issue. Here, we describe such a system based on conditional stability of essential proteins. We used a destabilizing domain degron stabilized by estradiol addition (ERdd). We ERdd-tagged 775 essential genes and screened for strains with estradiol dependent growth. Three genes, *SPC110*, *DIS3* and *RRP46*, were found to be particularly suitable targets. Respective strains showed no growth defect in the presence of estradiol and strong growth inhibition in its absence. *SPC110-ERdd* offered the most stringent containment, with an escape frequency of $<5\times10^{-7}$. Removal of its C-terminal domain decreased the escape frequency further to $<10^{-8}$. Being based on conditional protein stability, the presented approach is mechanistically orthogonal to previously reported genetic biocontainment systems.

Fueled by decreasing cost of DNA sequencing and synthesis, as well as an expanding toolkit for genetic manipulation of organisms from all kingdoms of life, synthetic biology is a rapidly advancing field. It holds promise to tackle key problems humanity is facing, e.g., by making agricultural and industrial production more efficient and sustainable, by providing bioremediation of environmental pollution or by offering novel solutions for previously intractable healthcare needs. Comprehensive safety engineering, a hallmark of every mature engineering technology, is required to safely deliver on this promise and pre-empt risks that could emanate from synthetic biology technologies and products.

One layer of biosafety engineering is the development and deployment of genetic biocontainment technologies. It is intended to curb an organism's ability to survive or transfer its genetic material outside of defined, permissive growth conditions. This intrinsic containment of genetically modified microorganisms is particularly attractive for large-scale or open-environment applications, or when working with pathogens. We have recently contributed an overview of approaches for genetic biocontainment of microorganisms[1].

Historically, genetic biocontainment work has been largely focused either on auxotrophies created by gene knockouts[2,3], or on control using different 'suicide genes', effectors like toxins or Cas endonucleases, with varying degrees of regulatory complexity[4–11]. Auxotrophic containment is generally genetically stable but has the caveat that the essential metabolite might be available in certain natural environments, e.g., by cross-feeding. In contrast, 'suicide' systems typically have a straightforward evolutionary trajectory for escape via the mutational inactivation of the effector gene. Due to these limitations of auxotrophic and suicide containment, control of essential genes for containment presents an attractive alternative approach. Genes essential under certain environmental conditions can be entirely knocked out, restricting survival in respective natural environments. For instance, removal of yeast's fluoride exporters has been used to make it sensitive to fluoride concentrations acceptable in drinking water[12]. Systems engineered to attain control over strictly essential genes may provide greater flexibility in choosing containment conditions. Such containment systems targeting essential genes along the flow of genetic information, from genomic presence[13] to

[1]Manchester Institute of Biotechnology, University of Manchester, Manchester, UK. ✉e-mail: yizhi.cai@manchester.ac.uk

transcription[13–15] and translation[15–18], have previously been developed. Combining different orthogonal control measures allows creating extremely stringent containment[13,15].

An approach so far not systematically explored for biocontainment is control over the in vivo half-life of essential proteins, adding post-translational control over essential genes. There are several switchable degron systems that allow targeted protein degradation in eukaryotes and have been used to link yeast cell viability to cues of light[19], temperature[20] or small molecules[21–23]. Control by small molecules absent in most natural environments seems to be an attractive approach to create versatile biocontainment systems. The auxin inducible degron system[21,22] and the SMASh tag[23] are small molecule-controlled switchable degrons that have been used in yeast on essential proteins, creating conditional cell fitness. Both are functional OFF switches, degrading the target protein in the presence of the molecule. However, this operation logic when applied to essential genes is not suitable for most biocontainment applications, in which proliferation in natural environments is to be prohibited.

Another class of switchable degrons functions as ON switch: the destabilizing domain (DD) degrons[24,25]. Here, we describe the systematic development of a highly stringent containment system based on an estrogen receptor-derived DD degron optimized for yeast (ERdd) (Fig. 1a). By screening more than 70% of *S. cerevisiae* essential genes, we have identified particularly suitable targets for biocontainment with this approach. Systematic characterization of containment escape modes allowed engineering an improved system with an escape frequency of <10[−8], an thus more stringent than optimized single-component transcriptional safeguards previously described in yeast[14]. We further demonstrated the feasibility to multiplex these protein switches and have constructed safeguarded strains with escape frequencies below the detection limit of the used assay ( < 2×10[−10]) that maintained wild-type-like fitness.

## Results

### ERdd system in yeast

The in vivo behavior of the yeast-adapted ERdd tag was tested by expressing a GFP-ERdd fusion protein from a centromeric vector over a range of estradiol concentrations and assessing GFP fluorescence (Fig. 1b) and abundance of the GPF fusion protein (Fig. 1c, Figure S1). In the absence of estradiol, specific GFP fluorescence was reduced by 95% compared to the fluorescence with 10 μM estradiol. At this highest tested estradiol concentration, specific GFP fluorescence of the GFP-ERdd samples reached 70% of that of untagged GFP. This indicated an exploitable dynamic window to leverage the ERdd tag to couple organism survival to estradiol supplementation by fusion to suitable essential proteins.

### Essential GFP-ERdd library creation and high-throughput screening for estradiol response

We performed a large-scale screen of essential genes for the desired growth response when fused to the ERdd tag—uncompromised fitness in the presence of 1 μM estradiol (permissive condition) and a severe fitness impediment in its absence (restrictive condition). To this end, the yeast GFP collection was leveraged to facilitate ERdd tag integration. The yeast GFP collection contains 4,159 strains, in which ORFs are individually tagged with a C-terminal GFP and a downstream *HIS3MX6* selectable marker gene. This constant region was used as a landing pad for C-terminal addition of the ERdd tag, allowing to use a single donor construct for editing of all strains of interest (Fig. 1d). The edit was mediated by a CRISPR/Cas9 system with an integration cassette also swapping the *HIS3MX6* marker for a *LEU2* marker gene, facilitating efficient editing and selection of correct edits. Out of 1103 *S. cerevisiae* ORFs deemed essential (https://rdrr.io/bioc/SLGI/man/essglist.html), 822 are physically represented as strains in the yeast GFP collection. Out of these, 775 strains (94%) were converted into GFP-ERdd fusion

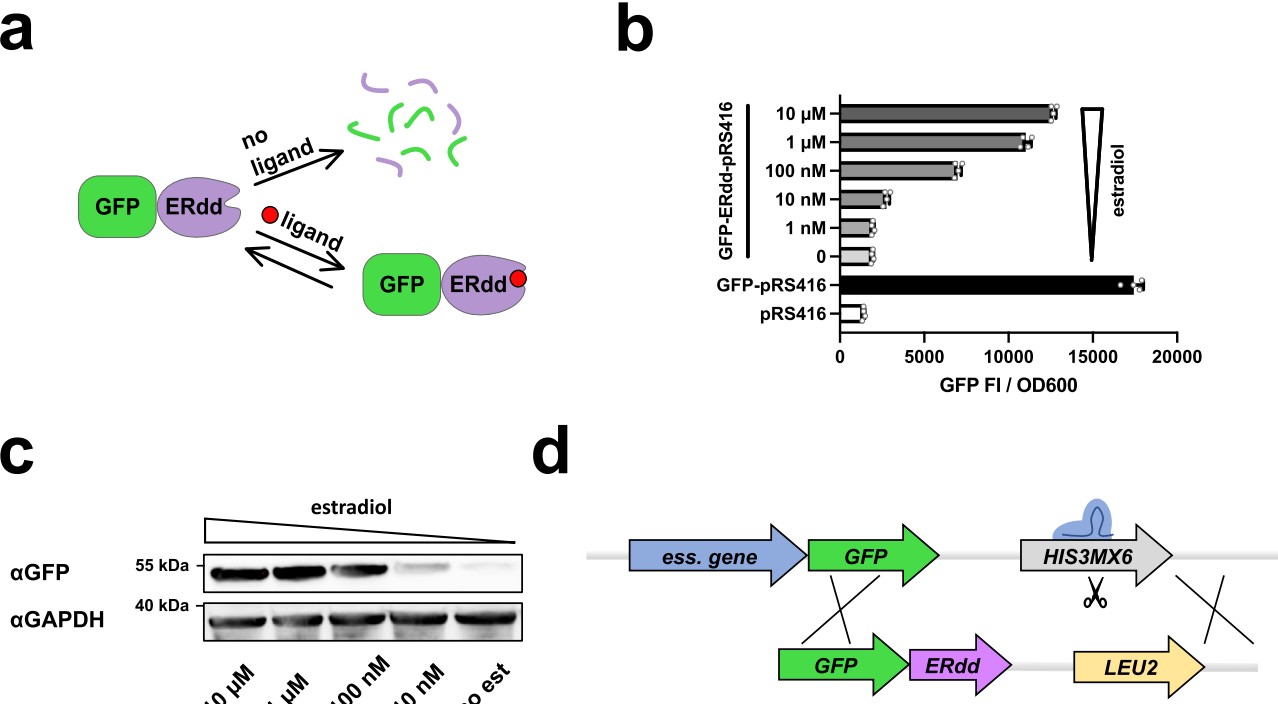

**Fig. 1 | ERdd tag for targeted protein degradation control in yeast. a** In the absence of its ligand, ERdd is in an unordered conformation, triggering its proteasomal degradation along with the protein it is fused to (here shown GFP). Availability of the ligand stabilizes ERdd and greatly increases half-life of the fusion protein. **b** Estradiol dependence of GFP fluorescence in yeast with a centromeric plasmid expressing a GFP-ERdd fusion protein. Shown are normalized fluorescence intensities from four replicates each, with individual data points shown as dots, means as center lines, and standard deviations as whiskers. Source data are provided as a Source Data file. **c** Western blot showing GFP-ERdd protein abundance is dependent on estradiol availability. This experiment was carried out a single time. **d** Scheme of Cas9-assisted conversion of essential GFP library to essential GFP-ERdd library with switch of marker gene from *HIS3MX6* to *LEU2*.

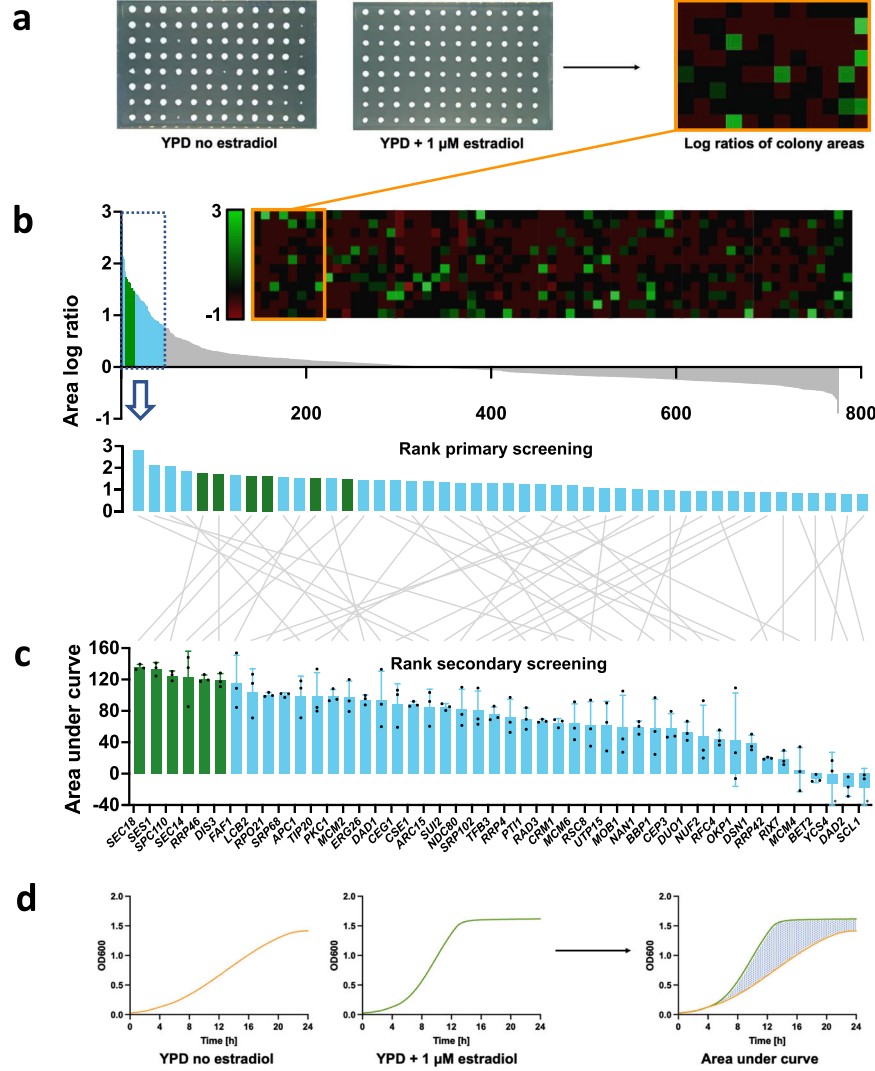

**Fig. 2 | Bipartite screening for estradiol dependence of essential GFP-ERdd strains. a** For the primary screening, essential GFP-ERdd strains were pinned on YPD agar without and with 1 μM estradiol. The area log ratios between permissive and restrictive condition were determined and used as metric for ranking. In the heatmap bright green indicates a strongly positive log ratio, i.e. a larger colony area with estradiol than without. **b** Measured area log ratios for the 775 strains in the primary screening were ranked from largest to smallest and the 46 highest-scoring strains were subjected to **c** the secondary screening in liquid culture. Individual data points are shown as dots, means as center lines, and standard deviations as whiskers from three biological replicates each. Asterisks at bottom whiskers of *YCS4* and *SCL1* each indicate a data point outside of the displayed range. Source data are provided as a Source Data file. The 6 highest-scoring strains in the secondary screening (marked in green) were created as direct fusions in a BY4742 background. **d** In the secondary screening, growth curves with and without estradiol were recorded for the 46 selected strains in three independent assays, and the mean area under the difference curve was used as metric for ranking.

strains (Supplementary Data S1). The others were either missing (24) in the library distribution we had, did not grow (4), or no Leu+/His-colonies could be obtained (19) upon transformation of the CRISPR/Cas components.

From the constructed essential GFP-ERdd library, estradiol dependent strains were identified by a bipartite screening (Supplementary Data S2). The primary screening for estradiol response was performed by pinning the generated essential GFP-ERdd library on YPD without (restrictive) and with 1 μM estradiol (permissive) and analyzing the colony size ratios (Fig. 2a) from a single experimental run. For most of the essential genes, ERdd-dependent degradation without estradiol appeared to not be substantial enough to cause a growth-inhibition phenotype, but about 5% of the screened strains showed a clear dependence on estradiol for proper growth (Fig. 2b). The 46 strains with the highest ratio of colony size between permissive and restrictive conditions were chosen for a subsequent screening in liquid culture in plate reader assays. For almost all of

these strains differential growth in response to estradiol was observed in the liquid culture screening as well (Fig. 2c, d). Strains were ranked by the mean area under the difference curve between growth under permissive and restrictive conditions from three independent experimental runs.

Most of the screened essential gene-*GFP-ERdd* fusions did not show a clear estradiol-dependent phenotype. To investigate whether this was due to a lack of ERdd-mediated essential protein degradation, the GFP-ERdd strains of the highly expressed essential genes *PGK1*, *YEF3*, *TPI1* and *GPM1* were assayed for growth and GFP fluorescence when cultured with and without estradiol (Figure S2). Relative fluorescence intensities suggested that substantial proportions of the respective fusion proteins were being degraded upon estradiol withdrawal (47–82% at 24 h). However, evidently the remaining essential protein amounts were sufficient to support growth, as the strains showed either no or only minor growth inhibition in the absence of estradiol.

**Estradiol response of single and double ERdd tagged strains**

The genes corresponding to the six highest ranked strains in the secondary screening, *SEC18*, *SES1*, *SPC110*, *SEC14*, *RRP46* and *DIS3*, were directly fused to ERdd in a BY4742 background without using any marker gene. The resulting six strains all showed a strong dependency on estradiol for growth (Fig. 3a, Figure S3). A genetic safeguard should not compromise the cell fitness under permissive conditions, as to not create an evolutionary incentive to inactivate the safeguard. Three of the strains, *SPC110-ERdd*, *RRP46-ERdd*, and *DIS3-ERdd*, exhibited a growth indistinguishable from the parental strain in permissive medium (YPD with $1\,\mu M$ estradiol) (Fig. 3b) and were used for further work. Of these, *SPC110-ERdd* displayed particularly favorable behavior, with a severe growth inhibition without estradiol and full restoration of growth already at 100 nM estradiol.

Subsequently, strains with two ERdd-tagged genes were created for combinations of *SPC110*, *RRP46* and *DIS3*. All dual ERdd strains exhibit strict dependence on estradiol. Growth of *SPC110-ERdd/RRP46-ERdd* was fully restored with 100 nM estradiol, whereas the other two strains required a higher estradiol concentration, reaching wild-type-like growth at $1\,\mu M$ estradiol (Fig. 4, Figure S4). After removal of estradiol dual ERdd strains initially underwent several doublings, but growth slowed and until it ceased altogether (Fig. 4c). In contrast, slow, but sustained growth of *RRP46-ERdd* and very slow growth of *DIS3-ERdd* could be observed in serial growth assays under restrictive conditions (Figure S5b).

**Escape rates and analysis of escapees**

The estradiol regulated strains not exhibiting any growth impairment under permissive conditions, *SPC110-ERdd*, *RRP46-ERdd*, *DIS3-ERdd* and derived double ERdd-tagged strains, were assessed for the stringency of containment they offer. This was done by assaying the frequency of escape colonies occurring on restrictive medium. To this end, eight independent cultures of the assayed strain were plated undiluted on YPD without estradiol (restrictive environment) and at a dilution of $10^{-5}$ on YPD with estradiol (permissive environment) to determine the number of colony-forming units.

Afforded containment varied drastically between the three genes when individually tagged with ERdd. *SPC110-ERdd* had the lowest escape frequency of $<3\times10^{-7}$, followed by *DIS3-ERdd* with $<4\times10^{-6}$, each over a period of 10 days (Fig. S6). *RRP46-ERdd* kept growing very slowly on restrictive plates, leading to the continued emergence of escaper colonies with prolonged incubation, making the quantification of an escape frequency infeasible. ERdd tagging both *RRP46* and *DIS3* yielded an escape frequency of $<3\times10^{-6}$ over 10 days (Figure S6c). Although not markedly improving escape frequencies over singly tagged *DIS3-ERdd* ($<4\times10^{-6}$) over the same timeframe, emergence of escapees appeared to plateau for the doubly ERdd-tagged strain after 6 days, but not for *DIS3-ERdd*. Interestingly, both *RRP46* and *DIS3* code for a subunit of the exosome complex essential for RNA metabolism.

Combining *SPC110-ERdd* with an ERdd tag on either *RRP46* or *DIS3* resulted in escape frequencies below the detection limit of this assay, not yielding a single escapee colony. Subsequently, assays for low escape frequencies were carried out for these strains, plating a total of about $5\times10^{9}$ colony forming units (CFUs) on restrictive plates, which were replica plated after two days for single colony detection and incubated for another 10 days. Again, no escapee colonies were observed for *SPC110-ERdd/DIS3-ERdd*. Accordingly, escape frequencies of this strain were below the assay's detection limit of about $2\times10^{-10}$ (Figure S7a). In contrast, escapee colonies for *SPC110-ERdd/RRP46-ERdd* started appearing after four days after replica plating. 10 days after replica plating an escape frequency of $1.9\times10^{-8}$ was determined, without yet having reached a plateau (Figure S7b). To validate the low escape frequency assay, it was carried out with *SPC110-ERdd/DIS3-ERdd* spiked $1:10^{8}$ with BY4742 for initial plating. Based on colony counts two days after replica plating, the assay yielded a mean frequency of $4.6\times10^{-9}$ (SD of $1.5\times10^{-9}$) viable cells, slightly underestimating the spiking frequency of $10^{-8}$.

Eight escapee strains from each of the three singly ERdd-tagged were isolated, and the ERdd-tagged gene was Sanger sequenced. For the sequenced *SPC110-ERdd* escapees, seven unique escape mutations were found, mapping to the C-terminal part of the Spc110 protein or the linker to the ERdd tag (Fig. 5). All of them result in a premature stop, leading the absence of the ERdd tag. Seven of the eight mutants were missing a C-terminal portion of the protein. The C-terminal region of Spc110p is known to be involved in binding to calmodulin, but not essential for viability[26]. Interestingly, no mutations within the ERdd tag itself were found, indicating some resistance against escape by truncation. Based on this finding, we generated an ERdd fusion omitting the non-essential C-terminal domain, *SPC110Δ845-ERdd*. This indeed increased containment stringency without compromising growth under permissive conditions, resulting in a strain with an estradiol dependence that was similarly favorable as that of *SPC110-ERdd*; growth of *SPC110Δ845-ERdd* likewise was fully restored at 100 nM estradiol (Fig. 5c). The strain's escape frequency was below the detection limit of the standard escape assay, not yielding escape colonies. In a low escape frequency assay over 10 days (after replica plating), an escape rate of $8.4\times10^{-9}$ was determined (Figure S7c). All but one of 19 sampled *SPC110Δ845-ERdd* escape mutations mapped to the linker or the N-terminal part of the ERdd tag, demonstrating that escape routes through truncating mutations in *SPC110* itself have been largely closed off by removal of its C-terminal region.

Based on the reasoning that N-terminal ERdd fusions would prevent escape through truncation mutations altogether, we tried creating those fusions for *SPC110*, *RRP46* and *DISS3*. *ERdd-SPC110* and *ERdd-DIS3* strains could be created and were assayed. Both were strongly dependent on estradiol for growth, but exhibited a modest, but clear fitness defect even at high estradiol concentrations (Figure S8). It is well known that N-terminal tags can lead to mislocalization of the fusion protein[27], which might be a possible reason for this observation.

The sampled *RRP46-ERdd*, *DIS3-ERdd*, and doubly tagged *RRP46-ERdd/DIS3-ERdd* escapees had no escape mutations in the respective fusion genes. To elucidate their escape mechanism, the whole genome of three clones of each were sequenced by Illumina sequencing (Supplementary Data S3). For each of the six escapees with an ERdd-tagged *DIS3* gene, a mutation likely to reduce proteasome activity could be found. Three unique mutations (two non-sense, one frameshift) of *RPN4*, a transcription factor stimulating expression of proteasome genes, were sampled. Further, a non-sense mutation of a chaperone for proteasome maturation (*UMP1*), and non-synonymous mutations of two proteins involved in proteasome regulation, *RPN3* and *RPT5*, were found. Likewise, for one of the *RRP46-ERdd* escapees a non-synonymous mutation of a proteasome subunit was the likely cause for containment escape. For the other two *RRP46-ERdd* escapees no obvious suppressor mutations could be identified.

**Competitive fitness of contained strains**

The perfect containment system imparts no fitness defect under permissive conditions as to not provide an evolutionary incentive to inactivate the containment system under prolonged culture. The strains contained by single or dual ERdd fusion to *SPC110*, *DIS3*, or *RRP46* and respective combinations showed growth that was indistinguishable from that of the parental BY4742 with $1\,\mu M$ estradiol in growth assays. However, a much more stringent measure for fitness is the competition against the parental strain under repeated, lab-typical batch cultures, cycling through different physiological states. Competition cultures were carried out over ten such cultures cycles for a total of 100 generations in YPD with $1\,\mu M$ estradiol (Fig. 6a), assessing the frequency of the parental and safeguarded strain at 0, 50 and 100

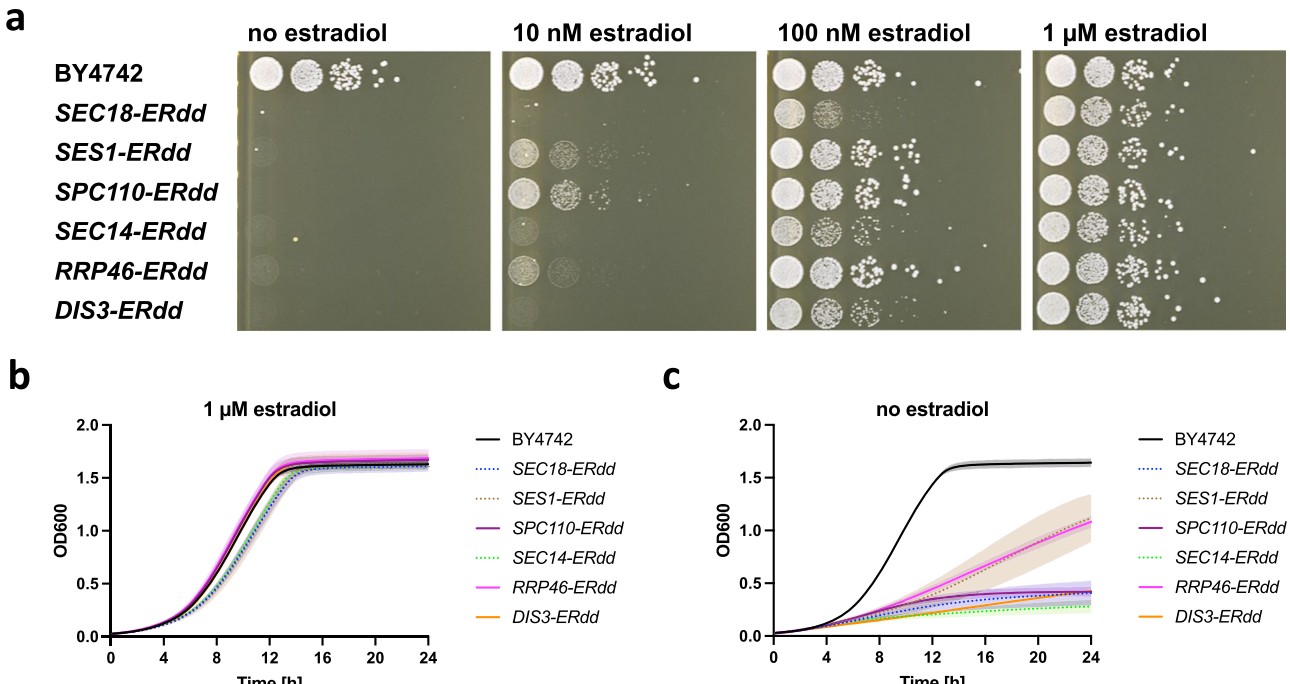

**Fig. 3 | Assaying growth of ERdd fusion strains for estradiol dependence. a** Spot assay of strains with direct ERdd fusion to a single essential gene at different estradiol concentrations on YPD agar, showing strong dependence on estradiol for all tested ERdd strains. **b** Growth assay in YPD with 1 μM estradiol, showing strains with growth indistinguishable from that of the parental strain BY4742 in solid lines and ones with a slight apparent growth defect with dashed lines. **c** All tested ERdd strains show a substantial to very strong growth defect without estradiol. For liquid growth assays means with standard deviations from three independent experimental repeats are shown. Source data are provided as a Source Data file.

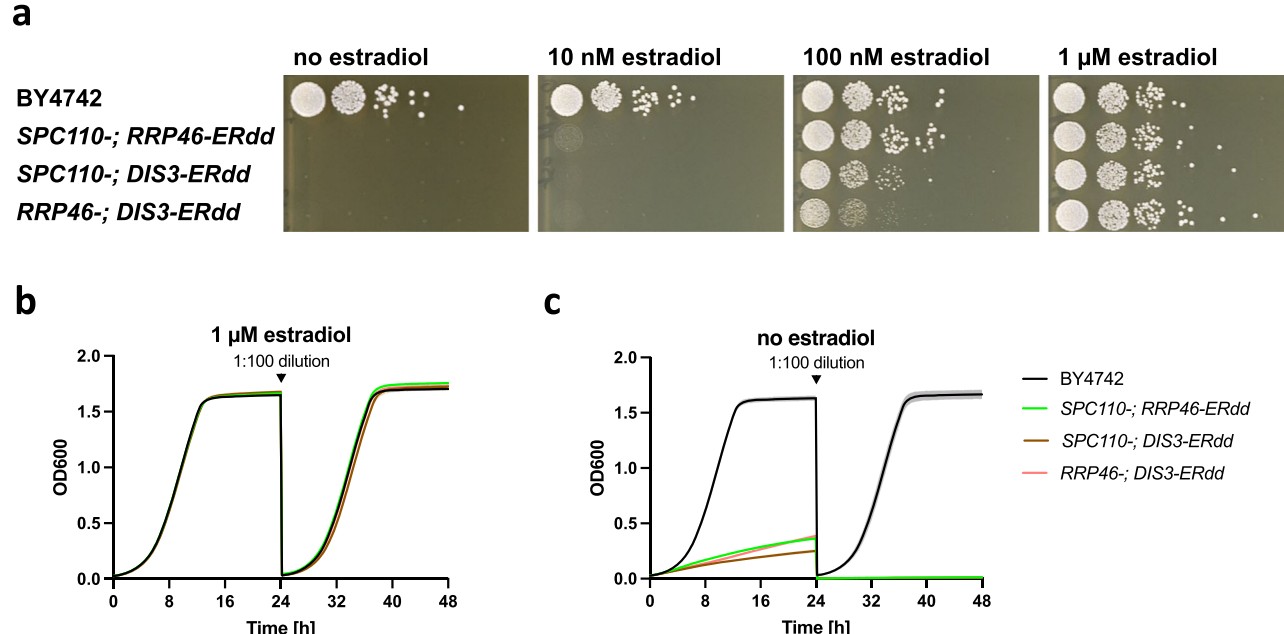

**Fig. 4 | Assaying estradiol dependence of strains with two essential genes fused to ERdd. a** Spot assay of strains at different estradiol concentrations on YPD agar, showing strict dependence on estradiol for all dual ERdd strains. Serial growth assays of dual ERdd strains in YPD **b** with 1 μM estradiol and **c** no estradiol, each showing means with standard deviations from five replicates. Source data are provided as a Source Data file.

generations from 46 to 48 samples each. None of the above-mentioned single and dual ERdd strains appeared to be outcompeted by the WT parental strain. This apparent lack of evolutionary incentive for containment inactivation should result in a high stability of the containment system in prolonged culture under permissive conditions. This was directly assessed by cultivation of eight independent *SPC110-ERdd* cultures over 100 generations under permissive conditions followed by assaying the escape frequencies. The measured average escape frequency after 100 generations was $2.2 \times 10^{-8}$ (2 days of incubation) not having increased following prolonged culture.

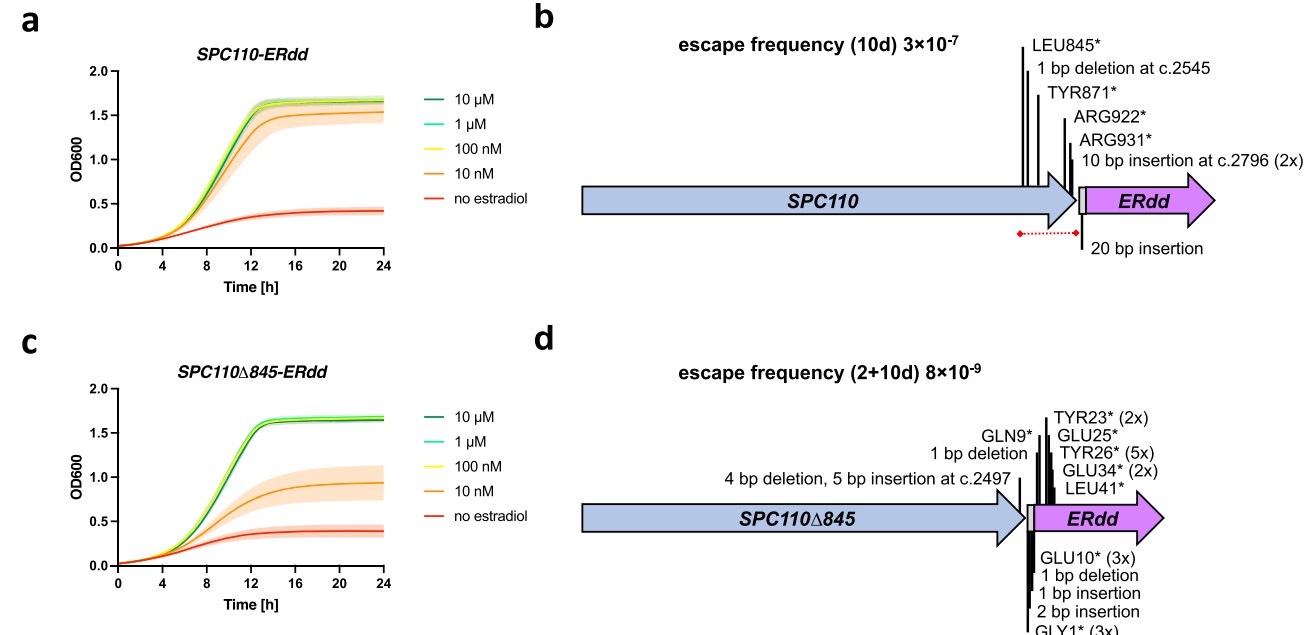

**Fig. 5 | Escapee analysis of *SPC110-ERdd* and engineering for improved containment stringency. a** Growth of *SPC110-ERdd* at different estradiol concentrations in YPD. **b** *SPC110-ERdd* containment escape mutations. Eight SPC110-ERdd escapees were sequenced, and seven escape mutations were found. All resulted in a premature translation stop by mutation to a stop codon or a frameshift, mapping to the C-terminal region of *SPC110* or the linker, and conversely removal of the ERdd tag. No mutations were found in ERdd itself. The dispensable C-terminal region of *SPC110* in which escape mutations clustered is marked with a red dashed line. **c** Growth of *SPC110Δ845-ERdd*, in which the C-terminal region has been removed, at different estradiol concentrations in YPD. As with *SPC110-ERdd*, growth is fully restored by 100 nM estradiol. Growth assays were performed with three biological replicates. Source data for a and c are provided as a Source Data file. **d** *SPC110Δ845-ERdd* containment stringency is markedly improved over *SPC110-ERdd*. 23 *SPC110Δ845-ERdd* escapees were sequenced. 14 separate escape mutations were found, with all but one of the sampled *SPC110Δ845-ERdd* escape mutations mapping to the linker or the N-terminal region of the ERdd tag. Residues are numbered separately for the *SPC110* coding sequence, the 10 amino acid linker (shown in gray), and the ERdd tag.

## Proteome and metabolome analysis of contained strains

To investigate effects of the introduction of ERdd tags on cell physiology under permissive conditions, cells grown in YPD with 1 μM estradiol were harvested in mid-log phase to analyze their proteomes by LC-MS. Normalized protein abundances of the safeguarded strains were compared to those of the parental strain. For each of the contained strains, the relative abundance of close to 3000 proteins could be assessed. Specifically, between 2939 and 2943 pairwise comparisons to the parental strain were made (Supplementary Data S6). For *SPC110-ERdd* no significant perturbations of the proteome were detected, whereas *SPC110Δ845-ERdd* (1 up-, 2 down-regulated), *RRP46-ERdd* (5 upregulated) and *DIS3-ERdd* (1 upregulated) showed apparent detectable changes of the proteome (Fig. 6b–e). Interestingly, in both *RRP46-ERdd* and *DIS3-ERdd* cytochrome b2 appeared to be upregulated. Both Rrp46p and Dis3p are components of the RNA exosome complex, but the relationship to cytochrome b2, a mitochondrial protein involved in lactate utilization, is unclear. To investigate a potential impact on the metabolome, cells were grown under the same permissive conditions as for proteome analysis and their metabolic profiles assessed (Supplementary Data S7). No significant differences between the metabolomic profiles could be detected by one-way ANOVA (FDR 0.05). Accordingly, in a principal component analysis, none of the assessed ERdd strains is distinguishable from the parental strain by the two principal components with the highest eigenvalues (Fig. S9).

## Discussion

We expect that a next-generation bioeconomy will aim to harness the potential of synthetic biology in open-environment applications. However, the safety of such applications hinges on the availability of stringent, stable and use case-suited biocontainment systems. Safely and effectively containing genetically engineered microorganisms through genetic systems poses a formidable challenge. The typically large numbers of individual organisms and their high replicative potential are problematic from a biocontainment perspective. An NIH guideline for work with synthetic or recombinant nucleic acids in laboratory settings calls for systems with an escape rate of less than $10^{-8}$ (https://osp.od.nih.gov/wp-content/uploads/2019_NIH_Guidelines.html). Since a single milliliter of a densely grown culture can easily contain $10^8$ cells, this threshold is clearly insufficient for applications on industrial scales. Reaching adequate containment stringency including a generous safety margin likely will require the combination of multiple independent systems.

Here, we have developed a class of genetic biocontainment systems based on conditional stability of essential proteins. Using a destabilizing domain (DD) conditional degron and applying a systematic, large-scale search for suitable essential genes from 775 genes, we have identified *SPC110*, *DIS3* and *RRP46* as suitable target genes for this system. Fusing *ERdd* to any of these genes rendered cells strictly dependent on estradiol without imposing a fitness defect when it is supplied. *SPC110-ERdd* was particularly suited, offering the most stringent containment under restrictive conditions of these three genes. Escapee analysis revealed that premature translation stops in the C-terminal region of Spc110p were the most frequent escape mechanism. Fusion of an accordingly truncated *SPC110* to the ERdd tag resulted in escape frequencies of $<10^{-8}$ over 10 days, satisfying the mentioned NIH guideline. Also, combining *SPC110-ERdd* with an ERdd tag on *DIS3* resulted in containment greatly exceeding these guideline requirements.

*SPC110-ERdd*, *SPC110Δ845-ERdd* and *SPC110-ERdd/RRP46-ERdd* required as little as 100 nM estradiol to achieve growth indistinguishable from the wild type. This equates to cost of less than $1 to treat 1000 L of culture, making it a highly cost-effective approach. The systems can be for instance combined with estradiol-dependent

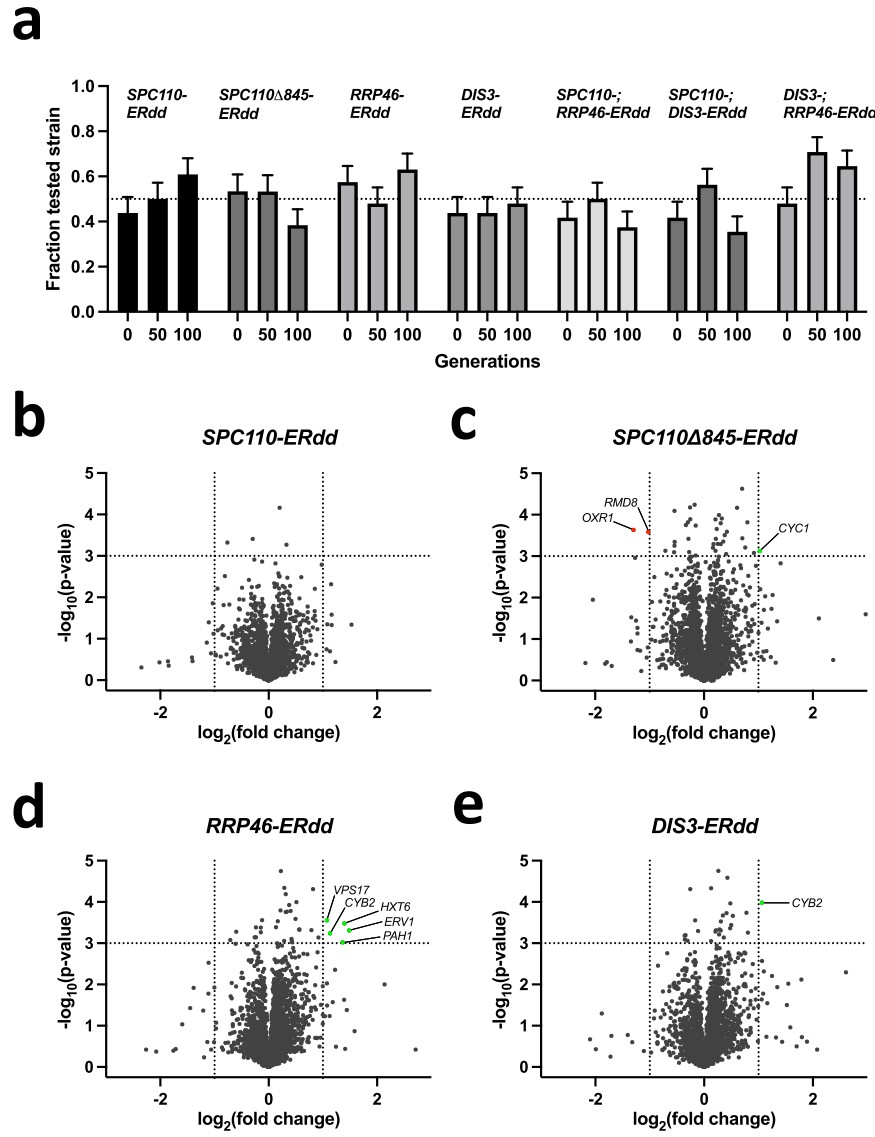

**Fig. 6 | Competitive fitness and physiological characterization of contained strains. a** Competition of singly and doubly ERdd-contained strains against BY4742 under permissive condition, in YPD with 1 μM estradiol. Respective contained strains were mixed 1:1 with BY4742 and grown for a total of 100 generations, with the relative frequencies of respective strains assessed after 0, 50 and 100 generations. Shown are calculated relative frequencies $freq_{rel}$ +/- SD of the binomial proportion estimation ($SD = \sqrt{freq_{rel}(1 - freq_{rel})/(n)}$) from genotyping independent colonies. Generally, the sample size $n$ was 48, but was 47 for *RRP46-ERdd* day 0 and 46 for *SPC110-ERdd* day 0 and *RRP46-ERdd* day 10. **b** to **e** show volcano plots of relative normalized protein abundances measured by LC-MS compared to the parental strain, using unadjusted *p*-values of two-sided t-tests of pairwise comparisons. Genes of up- and downregulated (*p*-value of <0.001 and a fold change of >2 or <0.5) proteins are labelled. Source data are provided as a Source Data file.

transcription of essential genes, which would allow controlling two independently working genetic containment systems with a single, near-gratuitous ligand at nanomolar concentrations. We have previously employed a chimeric, estradiol-inducible transcriptional activator of *GAL* promoters (GAL-EBD-VP16)[28] with select essential genes to create stringent biocontainment[13,14]. Alternatively, a more recent, similar system (ZEV)[29] allowing estradiol-dependent transcription from synthetic promoters could be used. Recently, a comprehensive screen of over 1000 essential genes with the ZEV system was reported, identifying several hundred genes with estradiol-dependent growth when under its control[30]. These could be purposefully screened for suitability for biocontainment.

The work presented here shows conclusively that conditional stability of select essential proteins can be leveraged to create stringent and stable biocontainment with specific ligand dependence. However, alternative small molecules might be beneficial use cases in which the utilization of estradiol for biocontainment is undesirable. For instance, large-scale use of estradiol with discharge into the environment can be environmentally problematic due to its endocrine activity[31]. Notably, there are more 'destabilizing domain' (DD) systems based on different protein scaffolds available that respond to other ligands: DD systems responding to the synthetic ligand Shield-1[24], trimethoprim[32], and bilirubin[33], respectively, have previously been developed. These alternative DD degrons should allow analogously creating biocontainment systems responding to their respective ligands.

## Methods

### Strains, plasmids, synthetic gene fragments and oligonucleotides
The yeast GFP collection[34] and the *S. cerevisiae* strain BY4742 were used for the creation of the essential GFP-ERdd library (Supplementary

Data S1) and generation of direct ERdd fusion strains (Table S1), respectively. DNA constructs made in this study as plasmids were cloned and amplified in *E. coli* DH5α. For creation of gRNA plasmids, pairs of oligonucleotides containing the respective target sequence and bases to create required overhangs (Table S2) were annealed. Subsequently, they were cloned into the gRNA entry vector pWS082 in a Golden Gate reaction (30 cycles of 1 min at 37 °C, 1 min at 16 °C and heat inactivation for 5 min at 80 °C) with 40 nM of annealed insert, 50 ng of pWS082, 5U Esp3I and 20U T4 DNA ligase in 10 µl T4 DNA ligase buffer. The *ERdd GFP LEU2* donor plasmid SHe146 (Supplementary Data S4) was cloned by Golden Gate assembly from parts PCR amplified from pScDD2_ERdd (Addgene plasmid #109047), pWS082 (Addgene plasmid #90516), genomic DNA from the yeast GFP collection, and pRS415[35]. Donors for direct ERdd fusions were ordered as gene fragments from Twist Bioscience (Supplementary Data S5). For confirmation of edits and for strain identification by colony PCR, pairs of primers spanning the end of the respective coding sequence were used (Table S2).

## Western blot

*S. cerevisiae* BY4742 was transformed with either GFP-ERdd-pRS416, GFP-pRS416 or empty pRS416. Transformed strains were grown overnight in Synthetic Complete with dropout of uracil (SC -Ura). The next day, fresh cultures in SC -Ura with different estradiol concentrations were inoculated 1:100 from overnight cultures and grown for 6 h at 30 °C. From each culture, the equivalent of 2 ml of a culture of an optical density of 1 was pelleted, washed with cold water and resuspended in 200 µl of 0.1 M NaOH and incubated for 10 min at room temperature. Cells were pelleted again and resuspended in 50 µl SDS-PAGE sample buffer (0.06 M Tris-HCl at pH6.8, 4% β-mercaptoethanol, 5% glycerol, 2% SDS and 0.0025% bromophenol blue). Samples were incubated at 95 °C for 5 min, and 20 µl of samples were run on precast 4-15% gradient PAGE gels (Mini-PROTEAN TGX Stain-Free, Bio-Rad) for 40 min at 200 V. Gels were blotted onto nitrocellulose with a Trans-Blot Turbo Transfer System (Bio-Rad). GFP was detected using anti-GFP GF28R (Thermo Fisher, 14-6674-82) at 0.5 µg/ml as primary antibody. As loading control GAPDH was detected using anti-GAPDH GA1R (antibodies.com, A85271) at 0.1 µg/ml. In both cases HRP-coupled rabbit anti-mouse IgG antibody (Sigma Aldrich, A9044) was used as secondary antibody at a 1:10,000 dilution.

## Creation of essential GFP-ERdd library

The yeast GFP collection (in a *S. cerevisiae* BY4741 background) was spotted in 384 well format on Synthetic Complete with dropout of histidine (SC -His) agar. The available GFP strains were cross-referenced with a list of *S. cerevisiae* essential genes (essglist). The respective strains with GFP labelled essential genes were cherrypicked onto SC -His agar in 96 well format using a Singer Instruments PIXL colony picking robot. The GFP cassette with the *HIS3MX6* marker gene was used as a landing pad for CRISPR/Cas9-based insertion of the ERdd as a C-terminal fusion to the GFP, while swapping *HIS3MX6* for a *LEU2* marker gene. For transformation, strains were inoculated into YPD medium in a deepwell plate and grown over night at 30 °C at 850 rpm. The next day, new cultures were set up in a deepwell plate by inoculating 1 ml YPD with 20 µl of the respective overnight culture each and grown for 4 h at 30 °C at 850 rpm. The cultures then were pelleted and washed with 1 ml, followed by another wash step with 0.5 ml 100 mM LiOAc. Supernatant was removed and a transformation mixture was added to achieve final concentrations of 33.3% PEG 3350, 100 mM LiOAc, 0.3 mg/ml boiled herring sperm DNA (Promega) with about 200 ng pWS174 digested with Esp3I (Cas9 plasmid), 400 ng of the *HIS3MX6* guide plasmid digested with EcoRV and 1300 ng SHe146 digested with NotI (*GFP-ERdd LEU2* donor). Digested plasmids were purified by isopropanol precipitation and ethanol wash prior to transformation. The deepwell transformation plate was thoroughly

vortexed and incubated 1 h at 37 °C at 850 rpm followed by an incubation at 42 °C for 20 min. Cell suspensions were pelleted, the supernatant removed, and the cells resuspended in 950 µl SC -Leu with 1 µM estradiol. These cultures selective for genomic insertion of the donor were incubated at 30 °C at 850 rpm. After 18 h, nourseothricin was added to a final concentration of 100 µg/ml to select for presence of the reconstituted Cas9 plasmid. After 48 h the transformation cultures were pinned onto SC -Leu agar with 1 µM estradiol using a Singer Instruments ROTOR replicating robot in a 7 by 7 pattern for each culture, creating a dilution series for each transformation outgrowth to obtain separated colonies. Up to four individual colonies were picked from each pinned transformation first onto SC -His, then onto SC -Leu and finally onto SC -Leu with 1 µM estradiol agar with the PIXL robot.

## Screening of GFP-ERdd library

The primary screening of the essential GFP-ERdd library for estradiol regulation was done on agar. From the SC -Leu +1 µM estradiol agar plates from the transformation, one (out of up to four) colony from each transformation was spotted with the PIXL robot, first onto YPD and then onto YPD + 1 µM estradiol agar in a 96 well layout. Only His auxotroph colonies were picked, and colonies with apparent regulation by estradiol were preferentially selected. The plates were incubated for 2 days at 30 °C and then imaged with a Singer Instruments Phenobooth, and colony sizes were assessed. The log ratio of the colony size with estradiol to the size without estradiol from a single experimental run was used to select the 46 highest scoring strains for the secondary screening.

The secondary screening was done in liquid culture in a plate reader. The strains selected from the primary screen were grown overnight in YPD with 1 µM estradiol at 30 °C and 850 rpm in a deep-well plate. The next day, the cultures were washed once with YPD without estradiol. In a 96 well F-bottom microtiter plate with lid, for each strain a sample of 200 µl YPD without and a sample of YPD with 1 µM estradiol was set up and inoculated 1:100 with the respective washed cell suspension. The assay was run for 24 h at 30 °C in a BioTek Synergy H1 plate reader, recording the optical density at 600 nm every 10 min while shaking between the measurement cycles. The growth curve without estradiol was subtracted from the growth curve with estradiol and the area under the difference curve was calculated. Strains were ranked by the mean area under the curve averaged from three assay runs.

To investigate degradation of essential GFP-ERdd fusion proteins, growth assays in a plate reader were carried out like the secondary screening, but in SC -Leu medium and with measurements of GFP fluorescence (excitation 479 nm, emission 520 nm, gain 100, with background subtraction) every 10 min in addition to optical density.

## Markerless ERdd insertion

The six highest ranked candidate genes from the secondary screening were created as direct markerless ERdd fusions in a BY4742 background using a CRISPR/Cas9 approach. A standard LiOAc yeast transformation protocol was used: exponentially growing yeast cells were harvested, washed first with water and then with 100 mM LiOAc. The transformation mix contained cells from 5 ml culture in 400 µl with final concentrations of 33.3% PEG 3350, 100 mM LiOAc, 0.3 mg/ml boiled herring sperm DNA (Promega) and about 500 ng pWS174 digested with Esp3I (Cas9 plasmid), 600 ng of the respective guide construct digested with EcoRV and 1000 ng of the respective donor DNA (Twist Bioscience). After thorough mixing, transformation cultures were incubated for 60 min at 30 °C at 200 rpm and then for 20 min at 42 °C. Cultures were spun down and resuspended in 500 µl YPD with 1 µM estradiol. After a transformation outgrowth period of 1-2 h at 30 °C under shaking, 100 µl were plated on YPD agar with 100 µg/ml nourseothricin and 1 µM estradiol and incubated for 3d at 30 °C. Colonies were picked with a PIXL robot onto YPD with 1 µM

estradiol. Insertion was confirmed by colony PCR. A correct clone was restreaked both on YPD with 1 μM estradiol, and loss of the Cas9 plasmid was confirmed by spotting colonies onto YPD with 1 μM estradiol and YPD with 1 μM estradiol and 100 μg/ml nourseothricin. Strains with two ERdd fusions were created the same way, starting with strains that already had one of the two genes ERdd tagged.

## Assaying estradiol response

The growth response of individual strains to estradiol was assessed both in liquid culture using a plate reader and in spot assays on solid media. Strains were cultured overnight in YPD with 1 μM estradiol, washed once in YPD, and adjusted to an OD 0.1 in YPD. With these cell suspensions, assays in the plate reader were carried out like the secondary screening, but at 10 μM, 1 μM, 100 nM, 10 nM and no estradiol. For spot assays, log10 dilution series of the OD 0.1 cell suspensions to OD $10^{-7}$ were prepared and 5 μl of each dilution were spotted on YPD agar at the different estradiol concentrations used (10 μM, 1 μM, 100 nM, 10 nM and no estradiol), incubated at 30 °C and imaged with a Phenobooth after two and three days.

## Escape rate analysis

For assessment of escape rates of ERdd strains, 8 individual colonies were picked into YPD with 1 μM estradiol and incubated overnight at 30 °C and 850 rpm. The next day the cultures were washed twice with YPD with no estradiol and a log10 dilution series to $10^{-5}$ was prepared. From each replicate 100 μl of the washed undiluted culture were plated on YPD agar without estradiol, and 100 μl of the $10^{-5}$ dilution were plated on YPD agar with 1 μM estradiol. Plates were incubated for a total of 10 days at 30 °C. The counted number of escapees on restrictive medium (no estradiol) after 2, 4, 6, and 10 days was divided by total colony forming units extrapolated from the colony number on the permissive medium (with estradiol) counted after 2 days. For each strain the average escape rate was determined from the 8 replicates.

For assessment of low escape frequencies ($< 10^{-8}$), 50 ml of overnight cultures (in YPD with 1 μM estradiol) were washed twice with water and resuspended in 5 ml YPD. This suspension was plated onto ten Petri dishes with YPD agar without estradiol. After 2 days at 30 °C, the dishes were replica plated onto fresh YPD agar without estradiol and incubated for another 2 days at 30 °C. To estimate the total plated number of colony-forming units, 100 μl of a $10^{-6}$ dilution of the washed and concentrated cell suspension were plated on permissive medium.

## Whole genome sequencing

Escapee strains and the parental strain BY4742 were grown overnight in permissive medium (YPD with 1 μM estradiol). DNA was isolated from 1.5 ml of culture with a MasterPure Yeast DNA Purification Kit and taken up in 50 μl TE buffer. To remove RNA contamination of the DNA preparation, 1 μl of 4 mg/ml RNase A (Promega) was added and the sample incubated for 30 min at 37 °C. The DNA was then column purified and eluted with 50 μl water. Sequencing library preparation, Illumina sequencing, mapping of reads and variant calling was performed by Eurofins Genomics. Reads were pre-processed with fastp v0.20.0[36] and mapped with BWA v0.7.17[37]. Subsequent variant calling was done using Sentieon's HaplotypeCaller. Variants in coding sequences that a) passed quality control filters, b) were not found in the parental strain and c) were present in at least half of respective reads were annotated using VariantAnnotation v1.40[38].

## Competition and stability assay

For assessing long-term competitive fitness compared with the parental strain, BY4742 and the ERdd strain to be assayed were each picked into YPD with 1 μM estradiol and incubated overnight at 30 °C. Overnight cultures adjusted to the same OD were mixed 1:1 and the mixture diluted 1:1000 in YPD with 1 μM estradiol and incubated for 24 h at 30 °C under shaking. Thereafter, the culture was rediluted

1:1000 for another 24 h of growth. This was repeated for a total of 10 days of growth of the mixed culture, representing about 100 generations. Aliquots of the initial mixture and the mixed culture after days 5 and 10 were plated on YPD agar with 1 μM estradiol to assess the genotype frequencies by colony PCR. To investigate long-term stability of the containment of *SPC110-ERdd*, the strain was similarly cultured over 10 days for about 100 generations in 8 independent cultures, from which an escape rate analysis was performed.

## Proteome analysis

Four replicates of containment strains and the parental strain BY4742 were grown overnight in permissive medium (YPD with 1 μM estradiol). Dense overnight cultures were diluted back 1:50 with YPD with 1 μM estradiol and grown for 4 h at 30 °C. From each culture, 1 ml samples were pelleted and washed twice with PBS. Cells were lysed in 5% SDS and 50 mM TEAB pH 7.5 using a Covaris LE220+ sonicator. Protein lysates were reduced, alkylated, acidified with phosphoric acid and the protein bound to wells of an S-Trap plate. After a tryptic digest, peptides were eluted and desalted using Corning FiltrEX desalt filter plates and OLIGO R3 beads. The liquid chromatography was performed on a Waters nanoEase M/Z Peptide CSH C18 Column using a Thermo RSCL system with a multistage gradient using 0.1% formic acid in water as buffer A and 0.1% formic acid in acetonitrile as buffer B with a runtime of 60 min. Mass spectrometry of the eluted peptides was carried out with a Thermo Exploris 480. Analysis of MS spectra was performed with Proteome Discoverer v2.5.0.400 using NCBI txid559292 v2022-04-30 (*Saccharomyces cerevisiae* S288C) as reference for identification of proteins. Proteins with at least two unique peptides were considered for quantitative comparison. For each analyzed contained strain, normalized abundances of found proteins were compared to the corresponding abundances in the parental strain. Proteins with a raw p-value of <0.001 and a fold change of >2 or <0.5 were considered significantly changed relative to the wild type. The mass spectrometry proteomics data have been deposited to the ProteomeXchange Consortium via the PRIDE[39] partner repository with the dataset identifier PXD042326.

## Metabolome analysis

Five replicates of each strain were grown overnight in permissive medium (YPD with 1 μM estradiol), diluted back 1:50 with YPD with 1 μM estradiol and grown for 4 h at 30 °C. 5 ml of each culture were pelleted and quenched with 10 ml of 60% methanol cooled to −70 °C. Cells were pelleted and resuspended in 1 ml of cooled 80% methanol. Cells were disrupted by snap freezing in liquid nitrogen, thawing on ice and vortexing for 30 seconds for three times. Samples were then centrifuged at 15,000xg for 5 min, the supernatant collected, and the solvent evaporated. All analysis was conducted on a Q Exactive Plus equipped with an Ultimate 3000 UHPLC (Thermo, UK). The UHPLC was equipped with an Accucore Vanquish C18+ reverse phase column (2.1 mm×100 mm; 1.5 mm particle size). The solvents employed were (A) water with 0.1% formic acid and (B) methanol with 0.1% formic acid. The flow gradient was programmed to equilibrate at 98% A for 1.5 min followed by a linear gradient to 2% A over 10 min and held at 2% A for 2 min before returning to 98% A for 1.5 min with a flow rate of 300 mL/min. The column was maintained at 45 °C and the samples chilled in the autosampler at 10 °C. A sample volume of 5 mL was injected onto the column. Blank injections were analyzed at the start and end of the analytical batch to assess the background and carryover. Data acquisition was conducted in full scan mode with data dependent acquisition for the top 5, in the scan range of 90-1350 m/z with a resolution of 70,000, an AGC target of $3e^6$ and a maximum integration time of 100 ms. Data dependent acquisition was undertaken with a resolution of 35,000, and AGC target of $1e^5$, a maximum integration time of 50 ms, a loop count of 5, isolation window of 4 m/z and a collision energy of 30. The acquisition was conducted in positive ion mode. Data analysis

was performed using MetaboAnalyst 5.0. Features with more than 20% missing values were excluded and the remaining missing values estimated by 1/5 of the minimum positive vale of each variable. For removal of near-constant variables, 20% of variables were filtered out based on the interquartile range. Samples were normalized by median, data $\log_{10}$ transformed and auto scaled. One-way ANOVA was performed with a 0.05 FDR cutoff.

## Statistics & reproducibility

No statistic sample size calculation was performed, and no prior assumptions about effect sizes were made. The primary screening for estradiol-dependent growth of the created GFP-ERdd library (775 strains) was done on semisolid media with a single experimental repeat. Strains were ranked by their log ratio between the respective colony sizes with estradiol and without estradiol. The more focused secondary screening of 46 of those strains was performed in three independent experiments in liquid culture, with a high degree of reproducibility for most strains. Growth curves with and without estradiol were recorded over 24 h for the 46 selected strains in three independent assays. Areas under the difference curves (with minus without estradiol) were calculated and averaged to be used as ranking metric.

For analysis of shotgun proteomics data, only proteins identified by at least two unique peptides were included. This pre-established criterion was chosen to ensure high confidence in the protein identity of analyzed abundances. No other data were excluded.

Growth assays were performed in at least three independent experiments each to ensure reproducibility. All attempts at replication were successful. No arbitrary experimental group allocation took place in this study. Experiments were not randomized and investigators were not blinded to allocation to experiments and outcome assessment.

## Reporting summary

Further information on research design is available in the Nature Portfolio Reporting Summary linked to this article.

## Data availability

Whole genome sequencing data has been deposited at SRA with the BioProject number PRJNA923142. Proteome data is available via ProteomeXchange with the identifier PXD042326. The Toronto BY4742 reference (available from https://downloads.yeastgenome.org/sequence/strains/BY4742/BY4742_Toronto_2012) was used for mapping, variant calling and annotation. For analysis of protein MS spectra in Proteome Discoverer, NCBI txid559292 v2022-04-30 (Saccharomyces cerevisiae S288C) (https://www.ncbi.nlm.nih.gov/Taxonomy/Browser/wwwtax.cgi?id=559292) was used. Yeast strains created for this work are available from Y.C. under a material transfer agreement. Source data are provided with this paper.

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

## Acknowledgements

This work is supported by grants from the EPSRC (EP/V05967X/1), the BBSRC (BB/W014483/1) and the Volkswagen Foundation "Life? Initiative" (Ref. 94,771), each awarded to Y.C. We thank Shuangying Jiang and Junbiao Dai for providing, and Ellie Payne and Reem Swidah for archiving the yeast GFP collection. We also thank Raymond Wan for variant annotation of whole genome sequencing data, Mark McCullough for help with NGS and proteomics data storage and deposition, and Elisa Barrow Molina for help with testing different degron systems. Further, we thank David Knight and colleagues at the University of Manchester BioMS core facility (RRID:SCR_020987), in particular Ronan O'Cualain for help with proteomics sample preparation, Stacey Warwood for running the samples, and Julian Selley for bioinformatic analysis. We thank the University of Manchester Faculty of Science & Engineering Mass Spectrometry and Separations Facility, in particular Katherine Hollywood, for metabolomics analysis. Finally, we want to thank Thomas Wandless and Rishi Rakhit for helpful discussions on the ERdd system. The Cas9 expression plasmid pWS174 (Addgene plasmid #90961) and the gRNA entry vector pWS082 (Addgene plasmid #90516) were gifts from Tom Ellis. The plasmid pScDD2_ERdd (Addgene plasmid #109047) was a gift from Thomas Wandless.

## Author contributions

Y.C. and S.A.H. conceptualized the study and acquired funding. S.A.H. designed and performed experiments, analyzed the data and wrote the original draft. Y.C. supervised the project, and reviewed and edited the manuscript.

## Competing interests

The authors declare no competing interests.
