## [Peer Review File · Nature Communications]

Reviewers' Comments:

Reviewer #2:

Remarks to the Author:

The rapid rise of the field of synthetic biology is enabling the increasingly sophisticated genomic engineering of bio-organisms. Alongside the potential benefits this can bring, there is the real concern about the risks associated with escape of the engineered organisms into the environment. Hence the engineering of in-built biocontainment measures into recombinant micro-organisms is key to safe progression from the controlled lab environment to applications in open environment settings.

In this manuscript Hoffmann and Cai present the development of such a biocontainment system for *S. Cerevisiae* yeast. The method is based on the drug-dependent stabilisation of proteins essential for yeast cell survival by their fusion to a degron that renders the fusion product unstable unless stabilised by the presence of the ligand, estradiol in this case. Screening a library of essential *S. Cerevisiae* genes they identify 3 proteins whose function, in terms of yeast cell growth, appears unaffected by the presence of the degron domain in its estradiol-stabilised conformation, but whose degradation in absence of the ligand causes efficient inhibition of cell growth.

This manuscript builds on previous work by one of the authors, but unlike that and some other 'kill switches', the system presented here has the beneficial feature that abrogation of growth occurs in the absence of the ligand; in other words, only when ligand is actively supplied do the engineered yeast cells proliferate. In that context, the choice for an estradiol inducible degron is perhaps somewhat unfortunate as nowadays estrogens are detectable in many locations in the environment, but as shown in the manuscript a relatively high estradiol concentration is needed for cell survival of the chosen ERdd tagged mutants.

The manuscript is well written and follows a logical progression of experiments.

There are a few points to consider:

1. The method is based on the fast and efficient proteasomal degradation of degron/destabilisation domain tagged proteins. However, throughout the manuscript protein degradation is only inferred, either from decrease in fluorescence levels in the case of the GFP tagged library or from reduction in growth, but there are no direct data to show differences in protein levels. Even though abolition of growth is the key objective from a practical viewpoint, it would be of interest to visualise what happens to the targeted proteins by Western blot. Degradation is assumed, but conformational changes due to the degron may also interfere with function.

2. For the selected ERdd tagged essential genes growth appears fully abolished in the absence of estradiol on the spot assays (eg. fig 3A, 4A). However, although strongly repressed, there is clear residual growth in liquid cultures. This issue is cleverly circumstepped in the text by focussing attention onto achieving normal growth when estradiol is added ('so as not to create an evolutionary incentive to inactivate the safeguard'), but clearly this will have to be addressed as it would be a serious impediment to the practical implementation of these potential safeguard strains.

3. In the section on analysis of the escapees the authors initially find premature stop mutations in the C-terminal portion of SPC110, but no mutations in the ERdd tag itself, leading them to claim that this 'indicated resistance of the ERdd tag against escape by truncation'. This would seem highly unlikely; it is much more likely that the observation is simply a spurious one due to the low number of escapees analysed. Indeed further analysis of the delta845 version revealed several ERdd tag mutations.

4. While the SPC110 escapees all have mutations in the gene itself, the RRP46 and DIS3 escapees appear instead to have mutations in proteasome associated genes. Some discussion of the reason for this difference would be appreciated.

5. The detailed analysis of the escapees is an important aspect of the study, but it highlights the potential for escapism by C-terminal truncations. The positioning of the degron on the C-terminus was obviously determined by the use of the GFP tagged library in the first screening, however in subsequent experiments it might have been a better choice to tag the degron onto the N-terminus, thus closing of the possibility of C-terminal truncating escape mutations. Has this been considered?

6. The paragraph on the proteome analysis of the contained strains mentions a small number of up- or downregulated proteins by number only. It might be useful to some readers to see what these proteins are, perhaps listed in a small table?

7. The first paragraph of the discussion starts with a couple of strangely convoluted sentences, which could surely be formulated better.

8. The discussion mentions that the NIH guideline for work with synthetic or recombinant nucleic acids in laboratory settings calls for systems with an escape rate of less than 10^{-8} . The escape rate of less than 2×10^{-10} measured here (in a spot assay; see point 2) is therefore impressive. Nevertheless, as pointed out by the authors in the discussion, 1 ml of culture can easily contain 10^8 cells and thus a relatively small volume of 100 µl could already have 10^{10} cells. The need for combinatorial safeguard systems seems clear.
9. A few minor points: In the 'Estradiol response of single ...' Section, figure 5A is referenced before figure 4 is mentioned.
10. In figure 4A the use of the ampersand in the double-tagged mutant names looks odd; a semi-colon might look better.
11. The colour has gone missing in Figure 5A and C, making it harder to distinguish the curves.

Reviewer #3:

Remarks to the Author:

The work by Hoffmann et al. employs an estradiol-inducible destabilizing domain (ERdd) to control the availability of essential proteins for the biocontainment of genetically engineered strains of *S. cerevisiae*. A comprehensive library of 775 ERdd-tagged essential genes, which could be an invaluable resource for genetic studies across disciplines, was generated to screen for essential genes that are compatible with ERdd. Finally, the authors managed to obtain final strains with stringent containment through ERdd-tagging of SPC110, RRP46, and DIS3, as well as their combinations. Overall, this study provides a novel strategy for biocontainment based on post-translational control of essential proteins, and the efficiency is competitive compared to existing strategies. However, further elaboration on the data is required, and there are concerns regarding the practicality of the method that needs to be justified.

- 1) There is a lack of citations of recent works addressing the same subject (for example: <https://doi.org/10.1038/s41467-020-19271-1>). The authors should elaborate on how their system is superior to existing biocontainment methods.
- 2) How many replicates were used to choose the essential genes in the primary screening? The use of colony size ratios as primary metrics for selection would be prone to errors if only one replicate were used. Please provide a more detailed description in the main text and methods section.
- 3) The escape rate of the system was evaluated by determining colony-forming units (CFUs) after two days of incubation at 30°C. Given this timeframe, it might be insufficient for slower-growing cells to form colonies, thus exaggerating the stringency of their system. Evidently, the liquid culture assays seem to suggest that even doubly-tagged cells could grow in liquid cultures (Figure 2C, 5A, 5C). Please provide further data showing CFU count at different incubation periods.
- 4) Page 11: "Between 2939 and 2943 protein abundances could be compared for each of the strains." Please improve the clarity of this sentence.
- 5) Page 11: "An NIH guideline for work with synthetic or recombinant nucleic acids in laboratory settings calls for systems with an escape rate of less than 10^{-8} ." A citation is needed for this claim.
- 6) Although the growth of the biocontainment strains is similar to the wild-type, it remains unclear if the performance of the biocontainment strains as production hosts would be compromised compared to the wild-type strain. Notably, the authors mentioned that a lactate utilization-related protein is upregulated. For completeness, the authors should conduct metabolomics analysis on the biocontainment strain.
- 7) Error bars for growth curves are missing in all figures. Please include the number of replicates in the figure legends.

We want to thank both referees for their careful and constructive review of our manuscript and their insightful comments and suggestions. We are particularly encouraged by their comments on the novelty and significance of this work: Reviewer 1 notes that “the system presented here has the beneficial feature that abrogation of growth occurs in the absence of the ligand” while reviewer 2 acknowledges that “Overall, this study provides a novel strategy for biocontainment based on post-translational control of essential proteins, and the efficiency is competitive compared to existing strategies.”

Based on their specific feedback, we have revised the manuscript as detailed here.

Reviewer 1

1.1 *The method is based on the fast and efficient proteasomal degradation of degnon/destabilisation domain tagged proteins. However, throughout the manuscript protein degradation is only inferred, either from decrease in fluorescence levels in the case of the GFP tagged library or from reduction in growth, but there are no direct data to show differences in protein levels. Even though abolition of growth is the key objective from a practical viewpoint, it would be of interest to visualise what happens to the targeted proteins by Western blot. Degradation is assumed, but conformational changes due to the degnon may also interfere with function.*

We thank the reviewer for this suggestion. We agree that a direct measurement of protein abundance to demonstrate fusion protein degradation is worthwhile. To address this, we have performed Western blot analysis on the GFP-ERdd construct. It shows clearly that the protein is being depleted in the absence of estradiol, and a dose-dependent rescue from degradation. We have added this data to Figure 1.

1.2 *For the selected ERdd tagged essential genes growth appears fully abolished in the absence of estradiol on the spot assays (eg. fig 3A, 4A). However, although strongly repressed, there is clear residual growth in liquid cultures. This issue is cleverly circumstepped in the text by focussing attention onto achieving normal growth when estradiol is added (‘so as not to create an evolutionary incentive to inactivate the safeguard’), but clearly this will have to be addressed as it would be a serious impediment to the practical implementation of these potential safeguard strains.*

For *RRP46-ERdd* residual slow growth without estradiol can be observed both in liquid culture and on agar in spot assays (Fig 3A) and when plating cultures. *DIS3-ERdd* appears to also show some sustained, albeit very slow, growth without estradiol. For the other strains, residual growth without estradiol seen in growth assays is only transient, and likely associated with the dynamics of target protein degradation or intracellular estradiol loss or dissociation from the degnon. To clearly demonstrate that residual growth is transient, we have performed sequential growth assays (now in Figure 4 and Supplementary Figure S4). A first growth assay of the contained strains was run (with and without estradiol). The cultures were then used to inoculate a second assay with the same conditions. This also dilutes any residual estradiol that would have been inside cells at the beginning of the first assay for the no estradiol samples. In the second assay, growth is absent in the restrictive condition for the strongly contained strains, i.e. *SPC110-ERdd* and doubly tagged strains. Some residual growth was seen for *RRP46-ERdd*, consistent with the observation in escape assays of continued slow growth. Very slow growth was also evident for *DIS3-ERdd*. To conclude, we have now demonstrated the

stringent containment capabilities in both solid agar and liquid culture conditions, and believe this addresses the reviewer's comment.

1.3 *In the section on analysis of the escapees the authors initially find premature stop mutations in the C-terminal portion of SPC110, but no mutations in the ERdd tag itself, leading them to claim that this 'indicated resistance of the ERdd tag against escape by truncation'. This would seem highly unlikely; it is much more likely that the observation is simply a spurious one due to the low number of escapees analysed. Indeed further analysis of the delta845 version revealed several ERdd tag mutations.*

Truncating escape mutations in the ERdd tag are indeed possible, as seen with the SPC110delta845 variant. However, the found truncation mutations were all clustered in a segment of less than 60 bp in *ERdd*, strongly suggesting that truncations at most other positions do not present viable escape routes. The distribution of found truncation mutations in *SPC110-ERdd* supports this: 7 of the 8 escape mutations mapped to the ~300 bp 3' region of *SPC110*, and the other to the 30 bp linker region, but none to the 730 bp of the ERdd tag. This observation was the basis for removing the C-terminal domain of SPC110, which we reasoned would close off a large portion of the sequence space of viable escape mutations. This evidently worked, as *SPC110delta845-ERdd* had a notably reduced escape frequency. Thus, we feel claiming that the ERdd tag is resistant to escape by truncation seems well justified. We have changed the wording to "indicating some resistance against escape by truncation" to avoid readers interpreting it as saying escape through truncation is not possible.

1.4 *While the SPC110 escapees all have mutations in the gene itself, the RRP46 and DIS3 escapees appear instead to have mutations in proteasome associated genes. Some discussion of the reason for this difference would be appreciated.*

We also found this difference notable and intriguing, as it appears to be the reason for the higher stringency of *SPC110-ERdd*. Here, mutational escape seems to require a nonsense or frameshift mutation in specific regions of a single gene. This makes it a low-probability event compared to an arbitrary loss-of-function mutation in any of several of genes.

Higher-order oligomerization of Spc110p is necessary to initiate assembly of the γ -tubulin ring complex essential for cell division (10.1091/mbc.E16-02-0072). This higher-order cooperativity of Spc110p may be the reason for a sharper cut-off between viable and non-viable levels of the essential protein compared to the other two essential target proteins Dis3p and Rrp46. We reason that viable (mutational) reduction of proteasome activity is not able to bring Spc110-ERdd levels above a minimal viable concentration threshold at which the protein is able to form homo-oligomers. However, other complex factors such as efficiency, processivity and kinetics of proteasomal degradation of the specific fusion proteins might also play a role. As such, in the absence of direct evidence, we feel it might be too speculative to include in the manuscript.

1.5 *The detailed analysis of the escapees is an important aspect of the study, but it highlights the potential for escapism by C-terminal truncations. The positioning of the degron on the C-terminus was obviously determined by the use of the GFP tagged library in the first screening, however in subsequent experiments it might have been a better choice to tag the degron onto the N-terminus, thus closing of the possibility of C-terminal truncating escape mutations. Has this been considered?*

Indeed, N-terminal ERdd fusions should be immune to truncation escape mutations and might thus create more stringent systems. However, we had not tried to N-terminally tag the winners of our screening previously. We had reasoned that creating an N-terminal instead of a C-terminal ERdd fusion would likely result in an altered estradiol response. However, encouraged by this comment, we decided to experimentally test this hypothesis. We have added a respective paragraph to the manuscript in the results section on containment escape. We tried ERdd-tagging the N-terminus of the winner genes *SPC110*, *RRP46* and *DIS3*. We managed to obtain strains with N-terminal ERdd fusions to *SPC110* and *DIS3*, but not to *RRP46*. For the latter, N-terminal fusions likely are not tolerated at all, or estradiol addition may not be able to rescue *ERdd-RRP46*. We assayed estradiol responses of *ERdd-SPC110* and *ERdd-DIS3* (now Supplementary Figure S6) and found both to be strongly dependent on estradiol addition, resembling the responses of the C-terminal fusions. The screening performed on C-terminal GFP-ERdd fusion strains offers some predictivity for desired behaviour of N-terminal ERdd fusions. However, neither of the two N-terminal ERdd strains recovered wild-type-like growth rates at the higher estradiol concentrations. As such, we reason they are not as suitable for genetic biocontainment.

1.6 *The paragraph on the proteome analysis of the contained strains mentions a small number of up- or downregulated proteins by number only. It might be useful to some readers to see what these proteins are, perhaps listed in a small table?*

The genes of up- and down-regulated proteins are labelled in the respective figures. We have added the respective cross-reference in the paragraph on proteome analysis.

1.7 *The first paragraph of the discussion starts with a couple of strangely convoluted sentences, which could surely be formulated better.*

Thanks for pointing this out. We have rephrased the beginning of the paragraph and hope to have improved its clarity: “We expect that a next-generation bioeconomy will aim to harness the potential of synthetic biology in open-environment applications. However, the safety of such applications hinges on the availability of stringent, stable and use case-suited biocontainment systems. Safely and effectively containing genetically engineered microorganisms through genetic systems poses a formidable challenge. The typically large numbers of individual organisms and their high replicative potential are problematic from a biocontainment perspective.”

1.8 *The discussion mentions that the NIH guideline for work with synthetic of recombinant nucleic acids in laboratory settings calls for systems with escape rate of less than 10 to the power of minus 8 . The escape rate of less than $2 \cdot 10$ to the minus 10 measured here (in a spot assay; see point 2) is therefore impressive. Nevertheless, as pointed out by the authors in the discussion, 1 ml of culture can easily contain 10 to the power 8 cells and thus a relatively small volume of 100 ml could already have 10 to the power 10 cells. The need for combinatorial safeguard systems seems clear.*

We fully agree with this assessment. Particularly when going beyond scales typically used in the lab, extreme stringency will be required. We believe multiple orthogonal, versatile, well-characterised safeguard systems are needed to fulfil this requirement.

1.9 *A few minor points: In the ‘Estradiol response of single ...’ Section, figure 5A is referenced before figure 4 is mentioned.*

We have removed this cross-reference to Figure 5A. The respective data is also presented in Figure S2, which is referenced in this section.

1.10 *In figure 4A the use of the ampersand in the double-tagged mutant names looks odd; a semi-colon might look better.*

As suggested, we have replaced ampersands with semicolons to indicate double-tagged strains in Figures 4 and 6.

1.11 *The colour has gone missing in Figure 5A and C, making it harder to distinguish the curves.*

We previously had used coloured curves only when displaying different genotypes and had presented series of curves from dose response data in greyscale. To make it easier for readers to distinguish curves and to facilitate including standard deviations, as reviewer 2 had suggested, we now use a coloured scale also for the dose response data.

Reviewer 2

2.1 *There is a lack of citations of recent works addressing the same subject (for example: <https://doi.org/10.1038/s41467-020-19271-1> [doi.org]). The authors should elaborate on how their system is superior to existing biocontainment methods.*

We have slightly expanded the introduction, including the suggested reference. As giving a comprehensive overview of the field is perhaps beyond the scope of this article, we also referenced a recent article of ours reviewing genetic biocontainment approaches.

Regarding it being superior to other systems – this seems like a strong claim, since usefulness of individual methods depends on the specific use case. However, the presented system evidently offers both high stringency and evolutionary stability, key benchmarks of genetic biocontainment systems. It makes survival dependent on near-gratuitous supplementation of a molecule not found in the required concentrations in natural environments and is thus highly versatile. Notably, it is based on a mechanism previously not exploited for genetic biocontainment. This orthogonality to existing approaches along with its good performance and high versatility makes it a valuable addition to the toolbox of containment systems, which we hope to have been able to convey in the manuscript.

2.2 *How many replicates were used to choose the essential genes in the primary screening? The use of colony size ratios as primary metrics for selection would be prone to errors if only one replicate were used. Please provide a more detailed description in the main text and methods section.*

The primary large-scale screen was performed as an individual run. Indeed, its prognostic accuracy might have been improved by performing more experimental repeats. However, this would likely not have improved the screening overall. The top regulated strains in the focussed secondary screening were all in the top tercile of the primary screen. The bottom hits from the primary screening showed little to none estradiol regulation in the focussed secondary screening. Both observations indicate that the screening was adequate in terms of balancing accuracy of the primary screen and size of the secondary screen to identify the best performers from the library.

The number of experimental runs is now mentioned in the main text and method section both for the primary and secondary screen.

2.3 *The escape rate of the system was evaluated by determining colony-forming units (CFUs) after two days of incubation at 30°C. Given this timeframe, it might be insufficient for slower-growing cells to form colonies, thus exaggerating the stringency of their system. Evidently, the liquid culture assays seem to suggest that even doubly-tagged cells could grow in liquid cultures (Figure 2C, 5A, 5C). Please provide further data showing CFU count at different incubation periods.*

This is a very good point. We have now performed escape assays over 10 days, and indeed additional escapees show up with prolonged culture, but interestingly to a very different extend. For *SPC110-ERdd* only few additional escapees emerged after 4 days. However, for singly tagged *DIS3-ERdd* there appeared to be a continued emergence of escapees, whereas escape frequencies of *RRP46-ERdd/DIS3-ERdd* only increased marginally after 4 days of incubation. This benefit of targeting both genes was not evident with the shorter escape assay performed previously. This data is supplied in Supplementary Figure S5 and described in the main body of the text.

Regarding growth in liquid culture please also see our response to reviewer 1's comment 2 and the serial growth assays performed to demonstrate that growth of biocontained strains in medium free of estradiol is mostly transient (see Figure 4 and Supplementary Figure S4).

2.4 *Page 11: "Between 2939 and 2943 protein abundances could be compared for each of the strains." Please improve the clarity of this sentence.*

To try to improve clarity of the statement, we have rephrased it to "For each of the contained strains, the relative abundance of close to 3000 proteins could be assessed. Specifically, between 2939 and 2943 pairwise comparisons to the parental strain were made."

2.5 *Page 11: "An NIH guideline for work with synthetic or recombinant nucleic acids in laboratory settings calls for systems with an escape rate of less than 10⁻⁸." A citation is needed for this claim.*

The mentioned guideline is now referenced in a footnote.

2.6 *Although the growth of the biocontainment strains is similar to the wild-type, it remains unclear if the performance of the biocontainment strains as production hosts would be compromised compared to the wild-type strain. Notably, the authors mentioned that a lactate utilization-related protein is upregulated. For completeness, the authors should conduct metabolomics analysis on the biocontainment strain.*

As suggested, we have performed metabolomic analysis. There is no indication of metabolic dysregulation as metabolomic profiles of the biocontainment strains are indistinguishable from the parental strain (see Supplementary Figure S7).

2.7 *Error bars for growth curves are missing in all figures. Please include the number of replicates in the figure legends.*

We have added standard deviations to growth curves as well as numbers of replicates in the respective figure legends.

Reviewers' Comments:

Reviewer #2:

Remarks to the Author:

Review of revised manuscript NCOMMS-23-22095A

Engineering stringent genetic biocontainment of yeast with a protein stability switch.

Stefan Hoffmann and Yizhi Cai.

In this revised manuscript Hoffmann and Cai present the development of a biocontainment system for *S. Cerevisiae* yeast based on the estradiol-dependent stabilisation of proteins essential for yeast cell survival, through fusion of these proteins to a degron that renders the fusion product unstable unless stabilised by the presence of the ligand, estradiol. Through screening of a library of yeast essential genes they have identified 3 proteins whose function, in terms of yeast cell growth, appears unaffected by the presence of the degron domain in its estradiol-stabilised conformation, but whose degradation in absence of the ligand causes efficient inhibition of cell growth. What makes these degron-controlled proteins suitable as potential genetic 'kill switches' is the feature that abrogation of growth occurs in the absence of the ligand; in other words, only when ligand is actively supplied do the engineered yeast cells proliferate.

In response to the reviewer's comments the authors have added some extra data and made small changes to the text, which have strengthened the manuscript.

I appreciate the addition of the Western blot showing a beautiful dose dependent stabilisation of the GFP-ERdd fusion product and its nearly complete absence in the absence of estradiol. While the more directly relevant fusion proteins to analyse by Western would be the SPC110, RRP46 and DIS3 fusions, I understand that, unlike widely available GFP antibodies, antibodies for these proteins will be harder and more expensive to obtain. Nevertheless, for the GFP-ERdd Western I would like to see the full blot, if not in figure 1C then at least in the supplementary data. Moreover, the M&M section mentions that samples were taken from GFP-ERdd-pRS416, GFP-pRS416 or empty yeast transformants. Figure 1C only shows the GFP-ERdd (in cropped blot format), but it would be good to see the blot with the controls in the supplementary data too.

While I remain somewhat sceptical about the claim that the ERdd tag is resistant against escape by truncation based on the low numbers of sequenced escapees in the study (it may well turn out to be so, but I would need a much larger number of mapped mutations to be convinced), it nevertheless appears beneficial to have the Spc110 C-terminal domain removed.

Reviewer #3:

Remarks to the Author:

The authors have made extensive revisions to address my comments. However, there are still some issues to address.

Methods, Escape rate analysis section: The manuscript's estimates of escape rates below 10^{-8} are contingent upon the effective transfer of cells from one YPD plate to another. However, there are currently no positive controls to confirm the efficiency of this transfer process and the sensitivity of the assay. It would be advisable for the authors to introduce a positive control, such as spike-in cells that are capable of growth on YPD plates in the absence of estradiol, within the SPC110-; RRP46-ERdd strain. This would enhance the credibility of the assay's sensitivity. Furthermore, a longer incubation time beyond two days should be performed to ensure a more reliable evaluation of the escape rates.

Again, we want to thank both reviewers for their careful examination of our manuscript. We are content that both reviewers appreciate the additional data and other changes (“*The authors have made extensive revisions to address my comments*”), and that they feel those additions and changes “*have strengthened the manuscript*”.

Reviewer 2

2.1 *I appreciate the addition of the Western blot showing a beautiful dose dependent stabilisation of the GFP-ERdd fusion product and its nearly complete absence in the absence of estradiol. While the more directly relevant fusion proteins to analyse by Western would be the SPC110, RRP46 and DIS3 fusions, I understand that, unlike widely available GFP antibodies, antibodies for these proteins will be harder and more expensive to obtain. Nevertheless, for the GFP-ERdd Western I would like to see the full blot, if not in figure 1C then at least in the supplementary data. Moreover, the M&M section mentions that samples were taken from GFP-ERdd-pRS416, GFP-pRS416 or empty yeast transformants. Figure 1C only shows the GFP-ERdd (in cropped blot format), but it would be good to see the blot with the controls in the supplementary data too.*

We agree, the Western blot of the GFP-ERdd fusion is a good addition and clearly demonstrates the mode of action of the ERdd tag is through protein degradation in the absence of ligand. The uncropped blots include the GFP-pRS416 and pRS416 controls and are now shown in Supplementary Figure S1.

2.2 *While I remain somewhat sceptical about the claim that the ERdd tag is resistant against escape by truncation based on the low numbers of sequenced escapees in the study (it may well turn out to be so, but I would need a much larger number of mapped mutations to be convinced), it nevertheless appears beneficial to have the Spc110 C-terminal domain removed.*

We have generated and sequenced 16 more *SPC110delta845-ERdd* escapees. Out of the total of 23 sequenced escapees, 13 have an escape mutation in the ERdd tag (see updated Figure 5D). All of these 13 mutations map to the first 41 amino acids of the tag (of 244), indicating that truncations further downstream do not present viable escape routes. This also explains the very high representation of the linker as target for *SPC110delta845-ERdd* escape mutations, with 9 of the sampled 23 mutations mapping to this short 30 bp stretch.

Reviewer 3

3.1 *Methods, Escape rate analysis section: The manuscript's estimates of escape rates below 10^{-8} are contingent upon the effective transfer of cells from one YPD plate to another. However, there are currently no positive controls to confirm the efficiency of this transfer process and the sensitivity of the assay. It would be advisable for the authors to introduce a positive control, such as spike-in cells that are capable of growth on YPD plates in the absence of estradiol, within the SPC110-; RRP46-ERdd strain. This would enhance the credibility of the assay's sensitivity. Furthermore, a longer incubation time beyond two days should be performed to ensure a more reliable evaluation of the escape rates.*

Both longer incubation times for the low frequency escape assays after replica plating and validating this assay by spiking with cells not requiring estradiol are good suggestions.

We have carried out the assay for the three strains in questions (*SPC110-*; *DIS3-ERdd*, *SPC110-*; *RRP46-ERdd*, and *SPC110delta845-ERdd*) with an incubation for 10 days after replica plating (see Figure S7). For *SPC110-*; *DIS3-ERdd* we still did not get any escape colonies, and for *SPC110delta845-ERdd* longer incubation had a very marginal effect (only yielding a single additional colony with prolonged culture). However, for *SPC110-*; *RRP46-ERdd* containment stringency notably deteriorated over prolonged culture (Figure S7B). All of the escapees showed escape mutations in *SPC110-ERdd*. Late occurrence of escapees might be linked to the observed very slow but sustained growth of the *RRP46-ERdd* strain in permissive media.

Secondly, we have performed the suggested positive control for this assay and added it to the manuscript as follows:

“To validate the low escape frequency assay, it was carried out with SPC110-ERdd/DIS3-ERdd spiked 10^{-8} with BY4742 for initial plating. Based on colony counts two days after replica plating, the assay yielded a mean frequency of 4.6×10^{-9} (SD of 1.5×10^{-9}) viable cells, slightly underestimating the spiking frequency of 10^{-8} .”

The results indicate that escape frequencies determined with this assay might be underestimating true values by a factor of about 2. As the reviewer suggested, this is likely due to losses from replica plating.